# Application of Crosslinked Polybenzimidazole-Poly(Vinyl Benzyl Chloride) Anion Exchange Membranes in Direct Ethanol Fuel Cells

**DOI:** 10.3390/membranes10110349

**Published:** 2020-11-17

**Authors:** Daniel Herranz, Roxana E. Coppola, Ricardo Escudero-Cid, Kerly Ochoa-Romero, Norma B. D’Accorso, Juan Carlos Pérez-Flores, Jesús Canales-Vázquez, Carlos Palacio, Graciela C. Abuin, Pilar Ocón

**Affiliations:** 1Departamento de Química Física Aplicada, Universidad Autónoma de Madrid, C/Francisco Tomás y Valiente 7, 28049 Madrid, Spain; ricardo.escudero@campusviu.es (R.E.-C.); kerly.ochoa@estudiante.uam.es (K.O.-R.); pilar.ocon@uam.es (P.O.); 2Instituto Nacional de Tecnología Industrial (INTI), Departamento de Almacenamiento de Energía, Av. General Paz 5445, San Martín B1650KNA, Argentina; rcoppola@inti.gob.ar (R.E.C.); gabuin@inti.gob.ar (G.C.A.); 3Departamento de Educación, Universidad Internacional de Valencia (VIU), C/Pintor Sorolla 21, 46002 Valencia, Spain; 4Departamento de Química Orgánica, Facultad de Ciencias Exactas y Naturales, Universidad de Buenos Aires, Buenos Aires C1428, Argentina; norma@qo.fcen.uba.ar; 5Centro de Investigaciones en Hidratos de Carbono (CIHIDECAR), CONICET—Universidad de Buenos Aires, Buenos Aires C1428, Argentina; 6Instituto de Investigación de Energías Renovables, Universidad de Castilla-La Mancha, Calle Investigación 1, Edif. 3, 02071 Albacete, Spain; JuanCarlos.PFlores@uclm.es (J.C.P.-F.); jesus.canales@uclm.es (J.C.-V.); 7Departamento de Física Aplicada, Universidad Autónoma de Madrid, C/Francisco Tomás y Valiente 7, 28049 Madrid, Spain; carlos.palacio@uam.es

**Keywords:** anion exchange membrane, polybenzimidazole, crosslinked, quaternized, fuel cell, direct ethanol

## Abstract

Crosslinked membranes have been synthesized by a casting process using polybenzimidazole (PBI) and poly(vinyl benzyl chloride) (PVBC). The membranes were quaternized with 1,4-diazabicyclo[2.2.2]octane (DABCO) to obtain fixed positive quaternary ammonium groups. XPS analysis has showed insights into the changes from crosslinked to quaternized membranes, demonstrating that the crosslinking reaction and the incorporation of DABCO have occurred, while the ^13^C-NMR corroborates the reaction of DABCO with PVBC only by one nitrogen atom. Mechanical properties were evaluated, obtaining maximum stress values around 72 MPa and 40 MPa for crosslinked and quaternized membranes, respectively. Resistance to oxidative media was also satisfactory and the membranes were evaluated in single direct ethanol fuel cell. PBI-c-PVBC/OH 1:2 membrane obtained 66 mW cm^−2^ peak power density, 25% higher than commercial PBI membranes, using 0.5 bar backpressure of pure O_2_ in the cathode and 1 mL min^−1^ KOH 2M EtOH 2 M aqueous solution in the anode. When the pressure was increased, the best performance was obtained by the same membrane, reaching 70 mW cm^−2^ peak power density at 2 bar O_2_ backpressure. Based on the characterization and single cell performance, PBI-c-PVBC/OH membranes are considered promising candidates as anion exchange electrolytes for direct ethanol fuel cells.

## 1. Introduction

The massive use of fossil fuels is partly responsible of climate change due to greenhouse gases emission into the environment. Urgently, it is necessary to develop new and cleaner energy-related technologies towards a more sustainable consumption model [1,2,3]. Electrolyzers and fuel cells are energy devices with high potential impact in such energy technologies [4,5]. For low temperature applications, among other alternatives, advantageous alkaline direct ethanol polymer electrolyte membrane fuel cells (DE-PEMFCs) and zero gap ion-exchange membrane electrolyzers can be used. They are comprised of positive and negative electrodes and an electrolyte based on a polymer membrane which plays both as an electrical insulator and ion conductor between the electrodes.

Some of the advantages of alkaline technologies are faster electrochemical kinetics, possible absence of noble metals as catalysts, minimized corrosion problems and cogeneration of electricity and valuable chemicals [6]. Ethanol is a non-toxic fuel that has the advantage of the liquid state, making it easy to handle, and can be considered a renewable fuel if it comes from renewable sources like the reformation of biomass residues.

The research of anion exchange membranes (AEMs), across which OH^−^ ions travel, is critical for the development of both technologies in order to reduce costs and widen their use [7,8,9,10], since a commercial, cheap, highly efficient and durable membrane has not been found yet.

Many efforts have been devoted to this purpose in recent years [6,11,12]. Positive quaternary ammonium head groups are still one of the main strategies to obtain high anionic conductivity through membranes, as shown in recent studies. Shaari et al. [13] used glycidyltrimethyl ammonium chloride (GTMAC) as the quaternizing agent on a poly(vinyl alcohol) (PVA) polymer backbone, obtaining 12.9 mS cm^−1^ at 30 °C, while Hao et al. [14] and Yu et al. [15] worked with a backbone of crosslinked polybenzimidazole-c-poly(vinylbenzylchloride) (PBI-c-PVBC) with the quaternizing agents N1-butyl substituted 1-butyl-4-aza-1-azaniabicyclo[2.2.2]octane bromide (BDABCO) and *N*,*N*,*N*’,*N*-tetramethyl-1,6-hexanediamine (TMHDA), respectively. They obtained 29.3 mS cm^−1^ at 20 °C using BDABCO and 31.5 mS cm^−1^ using TMHDA.

Polybenzimidazoles are well known for their excellent thermal and chemical stability [9] and thus have been extensively used for fuel cell membranes. Based on one of the most common members of the family, the poly [2-2′-(m-phenylene)-5-5′-bibenzimidazole] (PBI), and the crosslinking reaction with poly(vinylbenzylchloride) (PVBC), Lu et al. [16] synthesized various membranes and studied their application in a single H_2_/O_2_ fuel cell. Then they continued working with these types of membranes with various modifications and testing them for the same application [14,17]. They obtained a maximum peak power density of 230 mW cm^−2^ at 50 °C working with a PBI-c-PVBC membrane quaternized with 1,4-diazabicyclo[2.2.2]octane (DABCO) and the membrane presented good durability and stability [12]. Later, this result was even improved using 1-butyl-4-aza-1-azaniabicyclo[2.2.2]octane bromide (BDABCO) as quaternizing agent, which was added to the solution during the membrane crosslinking reaction. This procedure yielded a peak power density of 340 mW cm^−2^ with the M-BDABCO-OH-1:3 membrane at 50 °C [14].

In direct ethanol fuel cells (DEFCs), some of the highest performances have been obtained by Ma et al. [18], who reported a power density around 170 mW cm^−2^ with variations in the catalysts composition based on Pd and Ru or by Sun et al. [19], who achieved a power density of 202 mW cm^−2^ with anode catalyst of Pd nanoparticles on a carbon decorated porous Ni electrode, in both cases with a Tokuyama^®^ alkaline membrane electrolyte. Regarding membranes for alkaline DEFCs based on PBI, like KOH-doped ones, power densities ranging from 30 to 120 mW cm^−2^ at 60–80 °C are described in the literature [20,21,22].

Recently, we synthesized membranes of crosslinked polybenzimidazole-c- poly(vinylbenzyl chloride) (polybenzimidazole-c-PVBC) quaternized with 1,4-diazabicyclo[2.2.2]octane (DABCO), using PBI and poly(2,5-benzimidazole) (ABPBI), and tested them in a zero gap liquid alkaline water electrolyzer (LAWE) [23]. These membranes exhibit good stability in alkaline media, with ionic conductivity and current density values as high as 39 mS cm^−1^ at 25 °C and 380 mA cm^−2^ at a 1.98 V working at 50 °C, respectively. Based on these encouraging results, our research has been continued on PBI-c-PVBC membranes.

Thus, this work addresses a detailed study of the structural, morphological and electronic characteristics of the crosslinking and quaternization processes of these membranes, supported by XPS, SEM/EDX and solid ^13^C-NMR. Mechanical resistance and stability in oxidative media have been tested too, to evaluate their suitability prior to their use as alkaline electrolytes in DEFC. To the best of our knowledge, polybenzimidazole-c-PVBC crosslinked membranes had not been employed before in these types of fuel cells.

## 2. Experimental

### 2.1. Reagents and Preparation Procedure of the Membranes

Both the used reagents and the synthesis procedure are similar to our previous work [23]. Commercial PBI powder (BETWEEN, Lizenz GMBH Mw = 25,000–40,000 g mol^–1^), poly(vinyl benzyl chloride) (PVBC, 60/40 mixture of 3- and 4-isomers with average molecular weight Mw ~55.000 and Mw ~100.000, respectively, Sigma Aldrich, Merck Life Science S.L.U., Madrid, Spain), *N*,*N*-dimethylacetamide (DMAc, 99.5 wt.%, Sigma Aldrich), potassium hydroxide (KOH, 85 wt.%, Sigma Aldrich), ethanol (absolute, Sigma Aldrich), isopropanol (99.8 wt.%, Labbox Labware, Labbox Labware, S.L., Barcelona, Spain) and DABCO (99 wt.%, Sigma Aldrich), were purchased and used as received.

The membranes were prepared by solution of the polymers, crosslinking in solution at controlled temperature and casting of resulting solutions in Petri dishes to obtain the crosslinked membrane “PBI-c-PVBC”. The desired amount of solid PVBC was added to a 3.5 wt.% PBI solution in DMAc and stirred at 80 °C for 6 h in a dry closed glass vial. Finally, the mixture was placed in a Petri dish and the casting was performed in a vacuum oven (Memert VO200) at 40 °C and 100 mbar for 24 h to obtain the PBI-c-PVBC membranes. Afterwards, the membranes were quaternized by reaction with DABCO, immersing the membranes in 0.5 M DABCO solution in ethanol at 60 °C for 3 days, obtaining “Polybenzimidazole-c-PVBC/Cl” membranes, since the counter-ions of the formed quaternary ammonium groups were Cl^−^. Finally, ion exchange of Cl^−^ by OH^−^ was done by immersion in 1 M KOH aqueous solution for 2 days at room temperature, resulting in “Polybenzimidazole-c-PVBC/OH” membranes. Membranes of PBI-c-PVBC 1:1, 1:2 and 1:3 molar ratios were prepared using the corresponding PVBC amount.

### 2.2. Structure Characterization and Morphology

XPS measurements were performed in an ultra-high vacuum chamber at a base pressure better than 1 × 10^−9^ mbar. The XPS spectra were measured using a hemispherical analyzer (SPECS Phoibos 100 MCD-5, SPECS GmbH, Berlin, Germany). The pass energy was 9 eV yielding a constant resolution of 0.9 eV. The binding energies were calibrated using the Au 4f_7/2_, Ag 3d_5/2_ and Cu 2p_3/2_ lines of reference samples at 84.0, 368.3 and 932.7 eV, respectively. A twin anode (Mg and Al) X-ray source was operated using Mg Kα radiation (1253.6 eV) at a constant power of 300 W. The insulating character of the samples produced some charging effect, which was corrected by peaking the C 1s band attributed to C−C bonds at 285.0 eV and shifting accordingly all other core levels. Before introducing the samples in the system, they were dried at 60 °C under vacuum (10 mbar) for 3 h.

SEM microscopy was performed using a JEOL JSM-6490LV (JEOL, Tokyo, Japan) equipped with EDS (Oxford Link) and detector for backscattered electrons, operating at 20 kV. Membranes and membrane electrode assemblies (MEAs) were sputter coated with an Au-Pt target using an EMITECH coater.

The solid-state ^13^C CPMAS NMR spectra were recorded using a Bruker AV-400-WB spectrometer (Bruker Corporation, Billerica, MA, USA) at ambient temperature working at ^13^C resonance frequency of 100.61 MHz using zirconia rotors. Spin-lock cross polarization with magic angle sample spinning (CPMAS) was used to obtain the spectra and all samples were accumulated for 10 h. Chemical shifts were referred to tetramethylsilane (TMS, 0 ppm) and methylene group −CH_2_−) of adamantine (29.5 ppm) as primary and secondary references, respectively.

IR spectra were recorded in an FT-IR equipment (Perkin Elmer Spectrum Two, PerkinElmer España SL, Madrid, Spain) with attenuated total reflectance (ATR) accessory in the 1700 to 400 cm^−1^ range, with 2 cm^−1^ resolution and 16 scans per spectrum. Base-line correction was performed using the software of the equipment (Spectrum).

### 2.3. Mechanical Properties, Chemical Stability, Volume Swelling, IEC and Conductivity

Mechanical properties were evaluated by stress–strain measurements of the membranes. The tests were performed using a Zwick Z010 equipment (ZwickRoell, Ulm, Germany) with a 200 N static load cell. Shape and dimensions of the membranes are obtained according to ISO 37:2011 as dumb-bell test pieces (Type 4) with 2 mm width in the narrower portion. At least 3 pieces were stretched uniaxially for each sample at a constant crosshead speed of 10 mm/min until failure.

Oxidative stability of the membranes was investigated by reaction with Fenton’s reagent (aqueous solution of FeSO_4_ 4 × 10^−6^ M with H_2_O_2_ 3 vol.%). First, the membranes were soaked in distilled water for 24 h until complete hydration; then, they were soaked in Fenton’s reagent solution for 24 h at 40 °C in an oven. Next, they were rinsed with distilled water and the superficial liquid was removed with filter paper just before weighting them. Finally, they were introduced again in fresh Fenton’s reagent solution for another 24 h at 40 °C. This process was repeated for a total of 8 days of testing.

The membrane swelling was calculated based on the thickness, length and width before and after immersion of the dry membrane in 1 M KOH aqueous solution, at room temperature for 48 h.

The ion exchange capacity (IEC) determination was performed by titration with aqueous 0.02 M H_2_SO_4_ of 50 mL of 1 M NaCl aqueous solution where the membrane sample was immersed for 24 h.

The through-plane ionic conductivity measurements were determined by the impedance method (EIS), using an Autolab PGSTAT 30N (Metrohm Hispania, Madrid, Spain) coupled to a frequency response analyzer with a two-point technique.

### 2.4. Single-Cell Performance Test

A typical membrane–electrode assembly (MEA) was prepared with commercial carbon cloth as gas-diffusion layer (GDL) with a carbon microporous layer on one side (MPL) (ELAT - LT1400, Fuel Cell Store, 454 µm and 63% porosity) over which the deposition of the anode and cathode catalyst layers was performed. Catalysts Pt/C (Johnson Matthey, 40 wt.%) and PtRu/C (Johnson Matthey, 30 wt.% Pt and 15 wt.% Ru), for cathode and anode, respectively, were mixed with isopropanol, distilled water and Nafion^®^ ionomer (DuPont Nutrition and Biosciences Iberica S.L., Silla, Spain, 5 wt.% solution) to obtain a catalytic ink with a final weight ratio of solid compounds catalyst:Nafion^®^ of 96:4. The ink was sprayed onto the electrodes to reach 1 and 1.33 Pt mg cm^−2^ for cathode and anode, respectively. Previously obtained PBI-c-PVBC/OH membranes were sandwiched between anode and cathode, yielding MEAs with a 2.89 cm^2^ active area. Commercial pristine PBI membranes (M40 Dapozol^®^ membrane, 40 µm, Danish Power Systems Ltd, Kvistgaard, Denmark) used for comparison followed the same procedure, but they were doped in KOH 6 M for 5 days (considered the optimum doping time required for this membrane to achieve its maximum conductivity [24]) before their use. Arbin Fuel Cell Test System (Arbin Instruments, College Station, TX, USA) was used to produce the polarization curves. Pure O_2_ (Air Liquide) was fed at the cathode at a constant flow rate of 200 mL min^−1^ and different backpressures (0.5 to 3 bar). The anode was fed with a 2 M ethanol and 2 M KOH aqueous solution using an external peristaltic pump (Dinko Instruments, Barcelona, Spain) at a constant flow rate of 1 mL min^−1^. All experiments were carried out at 90 °C.

## 3. Results and Discussion

All membranes obtained according to the procedure detailed above showed very good homogeneity and flexibility, which made their handling easy (Figure 1).

The membranes show a light brown appearance, slightly darker as the PBI content increases, as shown by the decreasing color gradation for PBI-c-PVBC 1:1 to PBI-c-PVBC 1:3 (Figure 1a–c). During the crosslinking procedure and due to the entanglement of the polymeric chains, the viscosity of the solution continuously increases and it becomes a gel after approximately 12 h and cannot be handled properly, preventing it from being poured out of the flask into the Petri dish and therefore, stopping the casting process from being performed too (Figure 1d).

### 3.1. Structure Characterization and Morphology

#### 3.1.1. XPS

To provide information about the chemical reactions produced during the formation of the anion exchange membrane (AEM), and its subsequent quaternization with DABCO solutions, XPS has been used. Membranes of PBI-c-PVBC molar ratios 1:2 and 1:3 were analyzed.

Figure 2 shows the N 1s peaks of (a) pristine PBI, (b) pure DABCO, (c) PBI-c-PVBC 1:2 membrane and (d) PBI-c-PVBC/Cl 1:2 membrane after quaternization with DABCO. As can be observed, the pristine PBI membrane displays two peaks at 400.3 ± 0.3 and 398.7 ± 0.3 eV (Figure 2a) which are attributed to nitrogen of amine (–NH–) and imine (=N–) in the imidazole rings, respectively [24]. The ratio of imine to amine is close to one, as expected from the structure of the PBI molecule. DABCO compound (Figure 2b) displays a single broad peak at 402.9 eV attributed to nitrogen of NR_3_ group since both N atoms of the DABCO molecule are completely equivalent and therefore indistinguishable.

After the formation of the cross-linked membrane PBI-c-PVBC, Figure 2c, three peaks were observed at 398.7, 400.3 and 402.9 eV. It is worth noting that although the peaks at 400.3 and 398.7 eV are attributed to amine and imine nitrogen atoms of PBI, respectively, the ratio is now 0.6, therefore indicating that, at this stage, the membrane is not merely a physical mixture of PBI and PVBC but is a crosslinking product between PBI and PVBC. The new peak appearing at 402.9 eV is assigned to the nitrogen of the cross-linked species formed. The same peaks are observed in the PBI-c-PVBC 1:3 membrane (Appendix A), which is logical since they have similar structure and interactions between the polymeric chains.

We suggest two possible reaction mechanisms to explain the ratio modification after crosslinking process between the peaks at 400.3 eV (amine) and 398.7 eV (imine). The first is in agreement with Lu et al. [16]. It is based in the statement that the crosslinking reaction occurs via condensation between the methylene chloride group, CH_2_−Cl, and the secondary amine group, −NH−, of PBI. The peak at 400.3 eV (amine) would increase with respect to the peak at 398.7 eV (imine) due to the double crosslinking points formed to produce quaternary ammoniums. The quaternary ammoniums present a binding energy around 400.3 eV, indistinguishable from the amines. This value is in the range of others observed for quaternary ammoniums [25,26]. The new peak at 402.9 eV is assigned to the N of the crosslinking bond formed to yield a tertiary amine since the (−NH−) of PBI substitutes the H with the C from −CH_2_−Cl of PVBC. This is logical since the N on the tertiary amine of DABCO also appears at that binding energy.

Double crosslinking points are formed where an imidazole ring reacts with PVBC using both nitrogen atoms instead of just one (Figure 3c). Next to these positively charged groups there should be Cl^−^ counter-ions, but the Cl^−^ ion is not observed in the crosslinked membrane, only covalent Cl is identified, maybe because Cl^−^ ions are present in low amounts and thus are not detected.

The other mechanism to explain the results of the crosslinked membranes considers the crosslinking reaction taking place between the CH_2_-Cl groups of PVBC with the imine (=N−) groups in the imidazole rings of PBI. This would explain the reduction of the imine peak (398.7 eV) with respect to the amine one (400.3 eV). The new peak at 402.9 eV would be related with the formed N species. Since experimental data are consistent with both mechanisms, we cannot assure which is the prevalent one.

The effect of quaternization is analyzed from the results of Figure 2d and Appendix A, which show the N 1s of the PBI-c-PVBC 1:2 and 1:3 membranes, respectively, after treatment with DABCO. Again, three peaks are observed at 398.7, 400.3 and 402.9 eV. Now the ratio of the peaks at 398.7 and 400.3 eV is 0.09 after quaternization process and the peak of 402.9 eV increases its intensity relative to the others.

Since the experimental conditions of the quaternization step have similarities with those of the crosslinking process—extended periods of time at high temperature—it was suspected that the quaternization step could produce more crosslinking inside the membrane. However, this effect was not expected to be relevant due to the already conformed structure that would not allow a significant mobility of the polymeric chains. In order to investigate the possible influence, a procedure similar to the quaternization was performed in a crosslinked membrane, immersing it in EtOH at 60 °C for 3 days, but in the absence of DABCO. The XPS of the treated membrane after the procedure (Appendix A) was similar to the original crosslinked one, which means that the immersion of the membrane in ethanol at 60 °C for 3 days apparently did not change the internal structure. Thus, the changes observed in the quaternized membrane (Figure 2d) come solely from the presence and reactivity of DABCO.

When DABCO reacts with PVBC, the previous situation of two tertiary nitrogen atoms in the molecule changes to one tertiary nitrogen and one quaternary nitrogen. The last is the one now bonded to the methylene group, −CH_2_−, of PVBC.

The increase in intensity of the peak at 400.3 eV is ascribed to the higher amount of quaternary ammonium nitrogen atoms (coming from the N of DABCO bonded to PVBC). These nitrogen atoms present a binding energy that overlaps with the previous peaks at that energy, making them indistinguishable.

The intensity increase in the peak at 402.9 eV is due to the tertiary nitrogen atoms from the DABCO molecules, which do not react with PVBC. Therefore, the peak at 402.9 eV comes from the overlapping of this with that of the tertiary amines in the crosslinked membrane at 402.9 eV, in such a way that they cannot be accurately separated.

Regarding the chlorine atoms in the cross-linked membranes, the feature at 200.4 eV is assigned to the Cl 2p_3/2_ peak (Figure 3a) that is characteristic of covalent Cl species with a spin orbit splitting (*sos*) between Cl 2p_3/2_-Cl 2p_1/2_ of 1.6 eV. This result is in agreement with the fact that most of the Cl atoms remaining in the membrane after the crosslinking are in the unreacted CH_2_−Cl groups of PVBC, so they are covalently bonded. Moreover, the Cl 2p^3/2^ peak displays a shift from 200.4 to 198.8 eV from the non-quaternized membrane (Figure 3a) to the quaternized one (Figure 3b) revealing a change in the bonding character of Cl atoms from covalent to ionic upon quaternization with DABCO. This is in agreement with the fact that chloride ions are now counter-ions of the formed quaternary ammonium groups and no covalent chloride is detected, which confirms the complete quaternization of the CH_2_−Cl groups that avoided participation in the crosslinking step.

#### 3.1.2. SEM/EDX

Scanning electron microscopy (SEM) and energy dispersive X-ray analysis (EDX) were used to observe the superficial and cross-section microstructure of the membranes and MEAs and to monitor the chemical composition. Figure 4 and Appendix A show the cross-section and surface micrographs, respectively, of PBI-c-PVBC 1:2 and 1:3 membranes. Membranes exhibit a highly homogeneous structure with no apparent phase separation between the polymers.

EDX analysis of selected regions of crosslinked PBI-c-PVBC 1:2 and 1:3 membranes (Figure 4c,f) evidence the presence of Cl, mainly coming from the CH_2_−Cl bonds. The homogeneous distribution of Cl in the cross-section of the membrane confirms the presence of PVBC in the whole area and therefore, the success of the crosslinking reaction; a negative result regarding the homogeneity in the chlorine distribution would lead to localized lack of quaternary ammonium groups and therefore, impeded ionic conductivity. Moreover, a larger amount of Cl is observed in PBI-c-PVBC 1:3 in good agreement with the larger PVBC ratio.

A similar surface (Appendix A) and cross-section (Appendix A) analysis has been performed in the quaternized membranes, PBI-c-PVBC/Cl 1:2 and 1:3 leading to identical conclusions regarding homogeneity and Cl distribution. Therefore, according to their morphological and chemical characteristics, PBI-c-PVBC membranes do not change significantly after the quaternization process. Appendix A shows the N mapping through the samples. Again, the distribution corroborates the expected homogeneity in the membranes of the species containing N, PBI in the crosslinked specimens and also DABCO in those quaternized.

The SEM microanalysis of the quaternized membranes used in MEA configuration after being tested in the fuel cell rendered highly relevant results. Figure 5 shows micrographs of 1:2 and 1:3 PBI-c-PVBC/OH membranes, with no significant deformation or degradation, i.e., these membranes retain their microstructure and morphology after fuel cell tests. The grain-like structures observed in the PBI-c-PVBC/OH 1:2 membrane correspond to residual KOH.

The chloride mapping showed in Figure 5c,f presents the element distribution in the cross-section of both membranes that remains very homogeneous, as previously observed for former membranes. Additionally, the content of Cl in PBI-c-PVBC/OH membranes is very low, due to the OH^−^ exchange prior to their use as MEAs for fuel cell test, as showed in [23].

#### 3.1.3. Solid ^13^C-NMR

Due to the difficulty to dissolve the crosslinked membranes, solid ^13^C-NMR spectroscopy is a valuable technique to know about the chemical composition of crosslinked and quaternized membranes. Next, Figure 6 shows the spectra of PBI-c-PVBC and PBI-c-PVBC/Cl.

The peaks in the range of 100–160 ppm are assigned to the aromatic C atoms of PVBC and PBI and the imidazole rings of PBI, as it was described in the NMR results of PVA:polybenzimidazole membranes [27] and in literature [28,29]. Most significant changes due to quaternization appear in the δ = 30 to 80 ppm region. In the quaternized membrane (PBI-c-PVBC/Cl), the peak at 46.9 ppm, ascribed to the C bonded to Cl, vanishes, while two new peaks appear at 52.9 (C4) and 45.8 ppm (C5), ascribed to the C atoms of DABCO next to the quaternary ammonium and to the tertiary ammonium, respectively. This result confirms the quaternization of DABCO via one nitrogen atom, as it was intended. A new peak also appears at 68.1 ppm (C3), assigned to the C of PVBC bonded to the quaternary ammonium of DABCO instead of to the Cl atom, and thus shifted at a higher chemical shift. The C of PVBC bonded to PBI (C6) could not be easily identified as it would be sited in the broad band between δ = 30 and 60 ppm.

### 3.2. Mechanical Properties

Tensile tests of PBI-c-PVBC membranes crosslinked, before and after quaternization, were performed to determine their strength and evaluate the effect of quaternization in the mechanical properties of the membranes. Figure 7 shows the stress–strain curves of crosslinked (PBI-c-PVBC) and quaternized (PBI-c-PVBC/Cl) membranes obtained by stretching at a constant rate. According to their chemical form, crosslinked or quaternized, there are some differences in the mechanical behavior, although it can be merely described as a predominant elastic deformation region followed by a very weak transition to plastic deformation with no significant hardening, and necking during failure. Crosslinked membranes have a Young modulus of ≈1.5 and 1.2 GPa for PBI-c-PVBC 1:3 and 1:2, respectively, and close to 70 MPa for the ultimate strength. On the other hand, quaternized membranes exhibit 0.6 and 0.5 GPa values for the Young modulus of PBI-c-PVBC/Cl 1:3 and 1:2, respectively, while the ultimate strength is around 45 MPa in both cases. From these results, it is obvious that a significant decrease in the stiffness (Young modulus), yield strength and the maximum stress occurs upon the initiation of the quaternization process, whilst the plastic properties remain fairly similar. This points to a decrease in intermolecular forces due to the introduction of DABCO groups, which means less interactions of molecular chains after quaternization. From the inspection of the error bars in Figure 7b, a dispersion of the maximum stress values for most of the tested samples can be observed, which can be considered a normal circumstance for laboratory-synthesized membranes.

Comparing these results with those of PBI and previously developed PVA:PBI 4:1 membranes [24] (Figure 7b), the ultimate strength values of PBI-c-PVBC and PBI-c-PVBC/Cl membranes are in the same range as doped PBI and PVA:PBI membranes—52 and 36 MPa, respectively—but lower than pristine PBI—161 MPa [24]. As these membranes exhibit high fuel cell performance, we consider that the obtained mechanical properties for PBI-c-PVBC/Cl membranes are adequate for final device applications. Membranes were immersed in KOH 1 M at room temperature before the fuel cell test to convert them to PBI-c-PVBC/OH. Although the humidification process on quaternized membranes is expected to alter their mechanical properties (decrease in maximum stress and increase in elongation) [30], it should not be an obstacle for their use in fuel cells according to our previous experience with PVA:polybenzimidazole membranes in DEFC tests.

### 3.3. Degradation in Oxidative Media

Membranes to be used as electrolytes in fuel cells must be chemically stable in an oxidative media, since pure O_2_ is supplied constantly at the cathode side. Furthermore, products resulting from chemical reactions include H_2_O_2_, which in addition to its strong oxidant character can produce hydroxyl ions and radicals that attack the polymeric structure [31]. The Fenton test is a common tool to explore the suitability of membranes exposed to oxidative environment. The test was done in an aqueous solution with Fenton’s reagent (4 × 10^−6^ M FeSO_4_ and 3 wt.% of H_2_O_2_) at 40 °C, measuring the weight change of the samples each day and replacing the Fenton solution. It should be highlighted that the experimental conditions of this test are harder than that of a real fuel cell under operation and therefore it serves as an accelerated degradation test. Figure 8 shows the progressive loss of membrane mass as a function of time.

Pristine PBI membrane suffers a weight loss around 20% in 5 days [32], very similar to the degradation exhibited by PBI-c-PVBC/Cl membranes. At that time, the PBI-c-PVBC/Cl 1:3 membrane has suffered a weight loss of 24% while PBI-c-PVBC/Cl 1:2 exhibits only a decrease of 15%. Thus, the membranes show a very good stability in oxidative media, keeping similar or even better oxidative resistance than the pure PBI polymer. The results of PBI-c-PVBC/Cl membranes are consistent with the degree of crosslinking observed in gel fraction measurements [23], with gel fractions of 72.5% and 64.0% for 1:2 and 1:3 membranes, respectively. So, as the PBI-c-PVBC/Cl 1:3 membranes are less crosslinked, their skeletal structure chains are more accessible to chemical attack and therefore they suffer higher degradation compared to PBI-c-PVBC/Cl 1:2 ones.

The final appearance of the membranes after the Fenton test is shown in Figure 9. The PBI-c-PVBC/Cl 1:2 membrane shows the lowest degradation process, probably due to its higher resistance to the oxidative degradation.

Figure 10 and Appendix A show the FT-IR spectra of membranes before and after the Fenton test.

From the comparison of both spectra, the disappearance of the band at 1060 cm^−1^ wavenumber corresponding to the antisymmetric stretching vibrations of −NC_3_− group of DABCO molecules [33] clearly stands out, showing the complete degradation of the quaternary ammonium groups of the membrane under strong oxidant conditions. No other significant changes are observed in the IR spectrum compared with the parent membranes and therefore, its degradation by the oxidation process proceeds mainly by loss of the quaternized ammonium group. Considering the aggressive conditions of the experiment, the loss of this group and therefore also of the DABCO molecule, can be considered as a logical consequence of being the weakest bond of the membrane structure. However, it does not imply that this group is going to be degraded under fuel cell measurement conditions since these conditions are much milder than those of the oxidative degradation test.

With the presence of quaternary ammonium groups demonstrated and the good homogeneity and resistance (both to mechanical and oxidative media), previously illustrated, PBI-c-PVBC quaternized membranes are considered good candidates for the use in single direct ethanol fuel cells.

### 3.4. Performance in Alkaline Direct Ethanol Fuel Cell Test

Measurements of MEAs performance in the DMFC were developed for a 1 mL min^−1^ constant fuel rate of EtOH 2M/KOH 2M at the anode side and 200 mL min^−1^ of O_2_ gas at the cathode and 90 °C. Prior to the fuel cell test, membranes were doped and exchanged in KOH 1 M aqueous solution for 2 days and the pristine PBI membranes used as reference were doped for 5 days in KOH 6 M solution. The main difference between these two processes is that in the membranes with quaternary ammonium groups, the fixed positive charges retain the OH^−^ ions, apart from the contribution of OH^−^ ions introduced between the chains and interacting with PBI. Conversely, in pristine PBI, the absence of these positive fixed positions makes an effective incorporation and retention of the OH^−^ ions more difficult.

First, the membranes were measured at 0.5 bar backpressure and compared with a commercial membrane of PBI (Figure 11).

As can be observed, the PBI-c-PVBC/OH 1:3 membranes showed an open circuit voltage (OCP) of 750 mV and maximum power density of about 53 mW cm^−2^, very similar to commercial PBI membrane. This can be considered a good result since the homogeneity and the precision in the membrane preparation at the laboratory scale are not comparable to a commercial one. Moreover, the PBI-c-PVBC/OH 1:2 yielded a higher power density than previous ones, reaching up to 66 mW cm^−2^ at 337 mV, 25% higher than commercial PBI. The reason for the better performance of this membrane must be ascribed to an effective compromise between ionic conductivity and mechanical resistance. The higher OCP is related with a lower crossover of the reactants through the membrane from cathode to anode, with PBI-c-PVBC/OH 1:2 exhibiting an improved barrier effect and therefore reduced porosity and permeability compared to PBI and PBI-c-PVBC/OH 1:3. This effect can be attributed to the microstructure of the membrane, which leads to a better structural resistance.

In order to evaluate the resistance properties of membranes, the study of both mechanical and chemical degradation properties are valuable tools. Membranes of 1:2 and 1:3 ratios showed quite similar values regarding mechanical properties. However, the results of the accelerated degradation test in oxidative media were better for the 1:2 membrane: it was explained as a consequence of the higher crosslinking grade of the membrane (higher gel fraction). The mechanical resistance is a property relevant for these membranes under fuel cell measurement conditions, probably causing the difference in OCP between 1:2 and 1:3 ratios. It needs to be considered that the pressure imposed to the membranes in the single cell is perpendicular to the membrane surface, while in the mechanical test previously described the resistance is measured parallel to the membrane surface. As the mechanical resistance changes from one direction to the other, it may explain the different mechanical resistance behavior in the single cell test despite having similar maximum stress values. No cracks were observed in the cross-section membrane micrograph of the MEAs after being tested (Figure 5), meaning that the cross-over occurs through microfractures or micropores of a much smaller size.

PBI-c-PVBC/OH 1:2 and 1:3 have conductivity values at 90 °C of 30 and 45 mS cm^−1^ and ion exchange capacity (IEC) values of 1.74 and 1.97 mmol g^−1^, respectively (Table 1) [23]. The higher conductivity of the PBI-c-PVBC/OH 1:3 membrane is related with the higher density of quaternary ammonium groups, but it was not enough to compensate the cross-over through the membrane under fuel cell measurement conditions and thus the OCP and performance are lower than for PBI-c-PVBC/OH 1:2.

As is usually observed, higher O_2_ pressures yield better performance due to the improved permeation of O_2_ molecules in the winding microstructure of the cathode GDL and improved access to the catalytic sites. However, pressures too high lead to an increase in O_2_ crossover and the possibility to tear the membrane, with unavoidable mixing of reactants, including short-circuit events. To explore the limits of these membranes, we developed various tests with PBI-c-PVBC/OH 1:2 and 1:3 membranes at different O_2_ pressures (1, 2 and 3 bar). Since the 1:2 ratio showed better OCP and peak power density than 1:3 in previous fuel cell tests, it was interesting to test membranes of 1:1 ratio too, with the expectation of a higher mechanical resistance and OCP values due to the higher PBI content; consequently, these membranes were also synthesized and tested. Table 2 and Figure 12 show the results for the highest allowable O_2_ backpressure for each membrane.

According to O_2_ backpressure, the higher the PVBC content, the lower the maximum pressure that could be applied. Some membranes of 1:2 and 1:3 ratios could stand briefly at higher pressures, but most of them directly broke. The trend found with PVBC content is consistent since pure PBI membranes are much more resistant than those obtained by PBI doping or in combination with other polymers, as here with PVBC or other examples with PVA [24]. The best power density, 70 mW cm^−2^, was obtained for PBI-c-PVBC/OH 1:2 membrane at 2 bar backpressure. Overall, it was clear that membranes of 1:2 ratio presented better maximum power densities than those with 1:3 and 1:1 ratios, in this order. They also presented higher OCP values around ≈0.80 V, similar to those of 1:1 ratio and slightly better than the membranes of 1:3 ratio with ≈0.7 V. Therefore, although an increase in the backpressure (limited by the membrane mechanical resistance) could improve the reactants availability, other considerations regarding conductivity and crossover of the electrolyte must be considered, too.

Comparing with the state of the art in alkaline fuel cells fed with ethanol [34], the highest performances are around 150–170 mW cm^−2^ peak power density, with the best value of 176 mW cm^−2^ at 80 °C attained for an A201 membrane (Tokuyama^®^, Tokuyama Corporation, Tokyo, Japan) and Pd_3_Ru/C catalyst at the anode [18]. These results were obtained by optimizing the electrodes’ composition and using an industrial membrane electrolyte based on quaternized ammonium-containing polyolefinic aliphatic chains. Therefore, such experimental conditions and the compositions MEAs are far from the ones described in this research.

On the other hand, regarding PBI membranes as electrolytes, it is possible to find experimental conditions for alkaline DEFCs more similar to ours in other commercial developments. So, a 50 µm PBI membrane was measured using EtOH 2 M/KOH 2 M aqueous solution and pure oxygen in anode and cathode, respectively, at 90 °C and 2 bar backpressure [26]. The MEA configuration was Pt-Ru/C (30% Pt, 15% Ru) and Pt/C (20% Pt) for anode and cathode, respectively, that gives a very similar final catalysts loading to ours. They obtained a power density of 61 mW cm^−2^ and an OCP value of 0.98 V. According to this, although the OCP is higher, the maximum power density is slightly lower than ours (61 vs. 66 mW cm^−2^). This result can be considered as a significant consequence of the improved conductivity of PBI-c-PVBC/OH membranes with quaternary ammonium groups compared with that of the sole PBI chains. Thus, they provided a conductivity value of 18 mS cm^−1^ at room temperature, similar to ours at the same temperature [23], but lower than that obtained at 90 °C, 30 mS cm^−1^.

So, although PBI-c-PVBC/OH-based membranes are still far from certain industrial developments, their performance encourages us to improve and tune their properties in order to make them even more comparable to the current state of the art in anion conducting membranes. The effective crosslinking and the advantage of the quaternary ammonium groups within the structure lead to resistant, durable and highly conductive membranes that effectively retain the OH^−^ ions.

## 4. Conclusions

Crosslinked membranes were synthesized by reaction between PBI and PVBC in solution and a subsequent casting process in a vacuum oven. The obtained polymeric membranes were successfully quaternized with DABCO and exchange of Cl^−^ counter-ions by OH^−^ was performed previous to their final application.

XPS analysis showed the change in the ratio between the different nitrogen atoms in the crosslinked membranes relative to pure PBI due to the crosslinking reaction, where two different mechanisms are proposed. The incorporation of DABCO was detected both by the intensity increase in the associated peaks and the change in binding energy from covalent to ionic. The reaction of pendant CH_2_−Cl groups from PVBC in the crosslinked membrane with DABCO was also clearly observed by ^13^C-NMR, where the C atom of the methylene group suffered obvious changes and the signals of two different C atoms from reacted DABCO appeared.

The synthesized membranes presented a very homogeneous microstructure regarding the PBI and PVBC distribution and chemical composition based on the SEM/EDX results. While chemical stability under highly aggressive conditions was quite similar to PBI, mechanical resistance properties were lower, particularly for quaternized membranes. Regardless, both results suggest that these membranes are good candidates for direct ethanol fuel cells.

The single cell tests were performed with KOH 2 M EtOH 2 M aqueous solution in the anode and pure O_2_ in the cathode. The membranes’ performance was very good, improving the result of a commercial PBI membrane and was quite close to the best bibliography results under similar experimental conditions. The best performance was obtained with PBI-c-PVBC/OH 1:2 membrane, reaching peak power density of 70 mW cm^−2^ at 90 °C and 2 bar O_2_ backpressure.

## Figures and Tables

**Figure 1 membranes-10-00349-f001:**
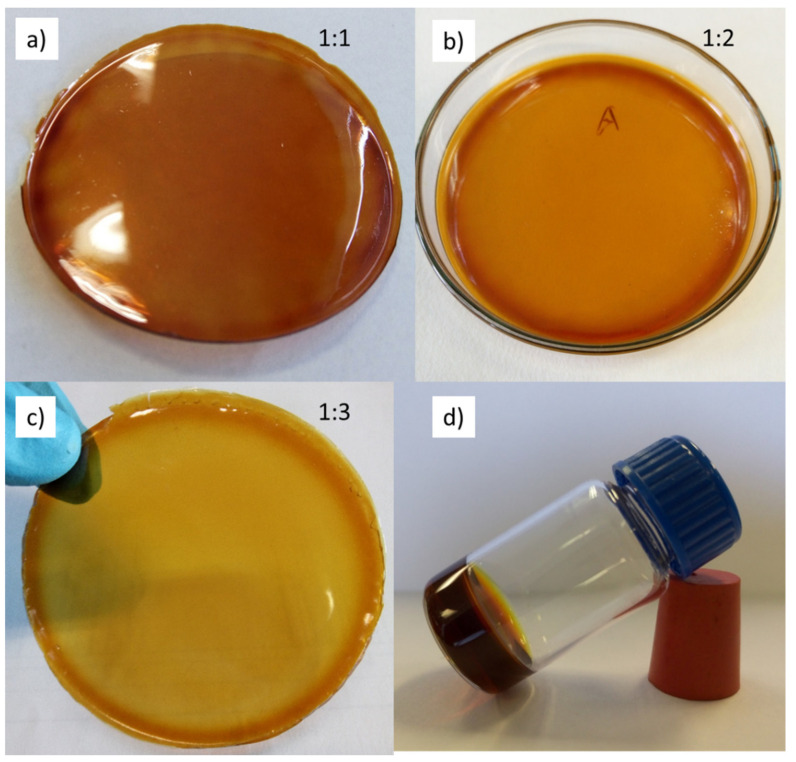
Aspect of PBI-c-PVBC membranes after 24 h of crosslinking with molar ratios (**a**) 1:1, (**b**) 1:2 and (**c**) 1:3. (**d**) Gelated mixture after 72 h of crosslinking at 80 °C.

**Figure 2 membranes-10-00349-f002:**
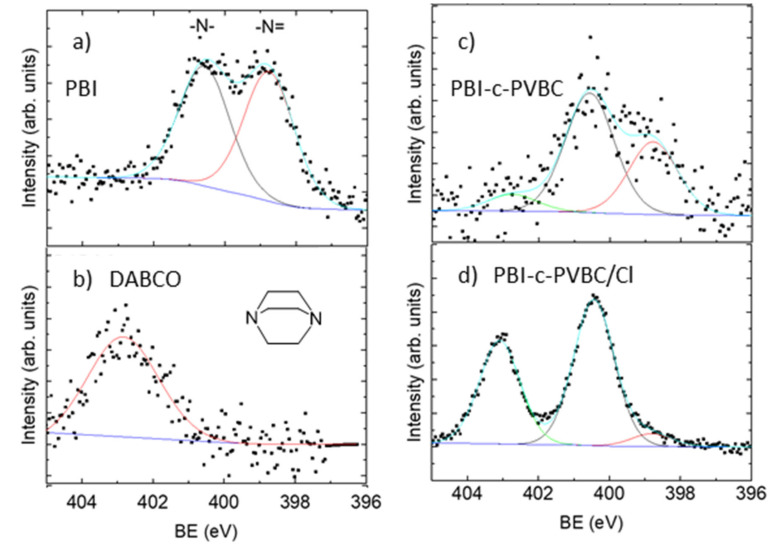
XPS N 1s peaks of (**a**) pristine polybenzimidazole (PBI), (**b**) pure DABCO, (**c**) PBI-c-PVBC 1:2 membrane (crosslinked) and (**d**) PBI-c-PVBC/Cl 1:2 membrane (after quaternization with DABCO).

**Figure 3 membranes-10-00349-f003:**
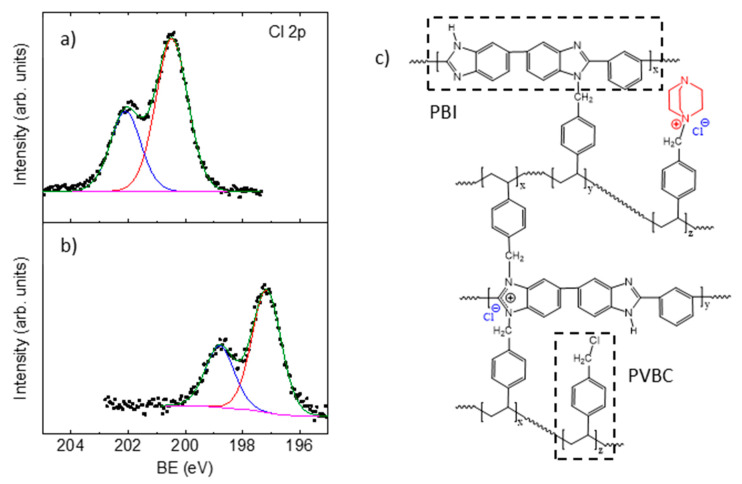
XPS Cl 2p doublet of (**a**) PBI-c-PVBC 1:2 membrane (crosslinked), (**b**) PBI-c-PVBC/Cl 1:2 membrane (after quaternization with DABCO) and (**c**) schematic chemical structure of the membranes.

**Figure 4 membranes-10-00349-f004:**
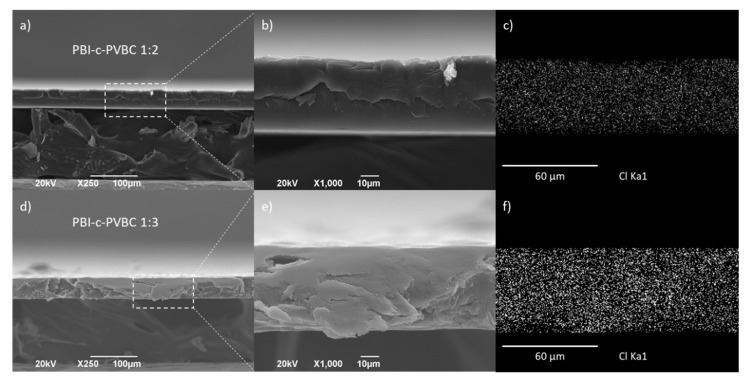
Cross-section (**a**,**d**) SEM micrographs; (**b**,**e**) detailed inset and (**c**,**f**) corresponding Cl EDX mapping of PBI-c-PVBC 1:2 and 1:3 membranes, respectively.

**Figure 5 membranes-10-00349-f005:**
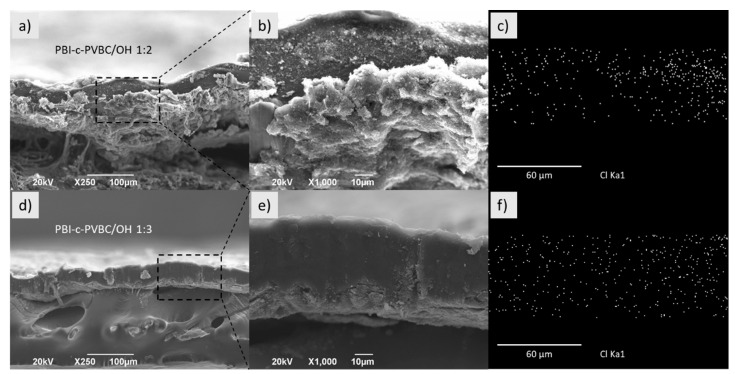
Cross-section (**a**,**d**) SEM micrographs; (**b**,**e**) detailed inset and (**c**,**f**) corresponding Cl EDX mapping of PBI-c-PVBC/OH 1:2 and 1:3 membranes, respectively.

**Figure 6 membranes-10-00349-f006:**
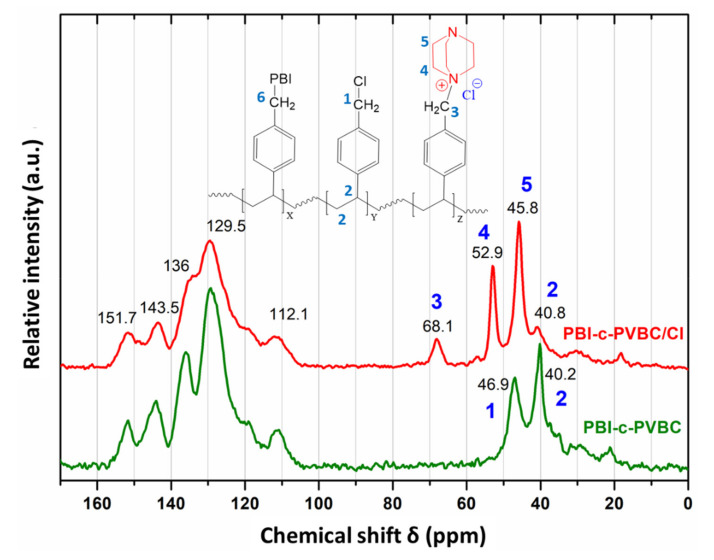
Solid ^13^C-NMR spectra of PBI-c-PVBC and PBI-c-PVBC/Cl.

**Figure 7 membranes-10-00349-f007:**
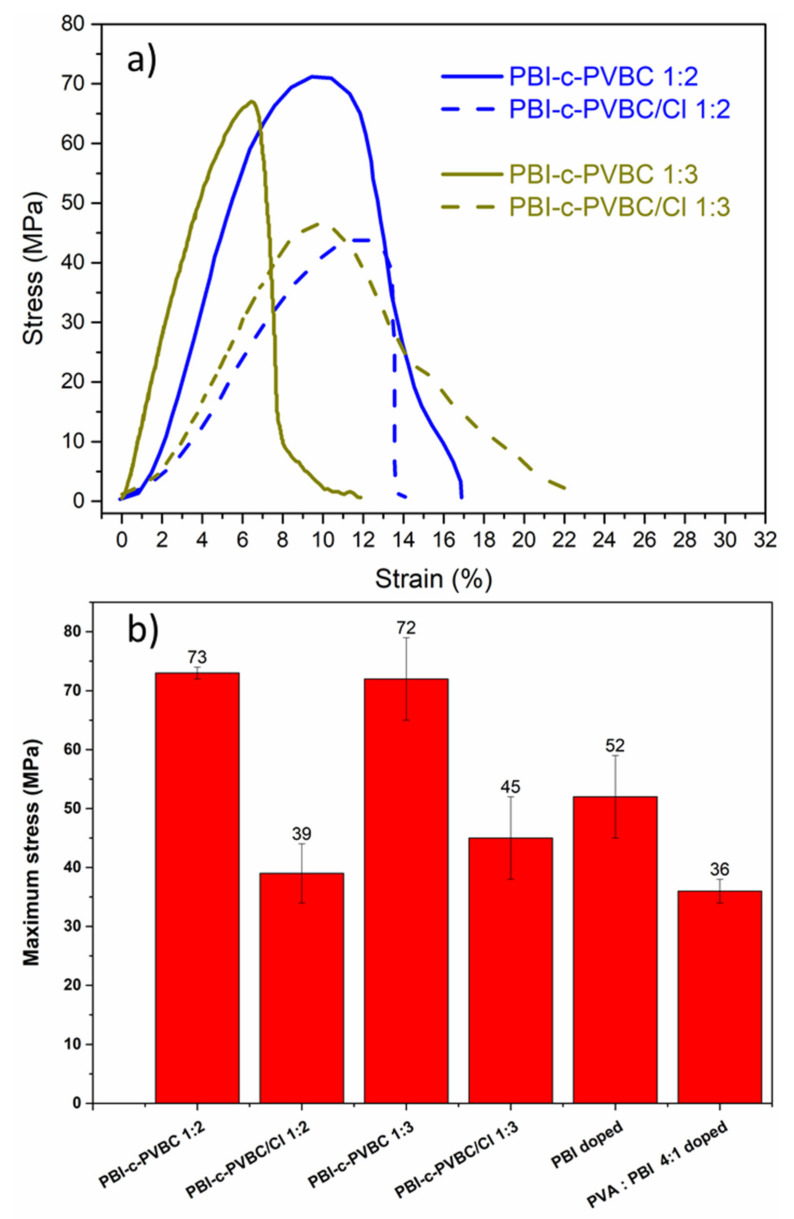
(**a**) Tensile strength results (average curve) of the crosslinked and quaternized PBI-c-PVBC membranes and (**b**) maximum stress values of the previous membranes compared with PBI and L-PVA:PBI 4:1 membranes in KOH 6 M for 5 and 3 days, respectively [24].

**Figure 8 membranes-10-00349-f008:**
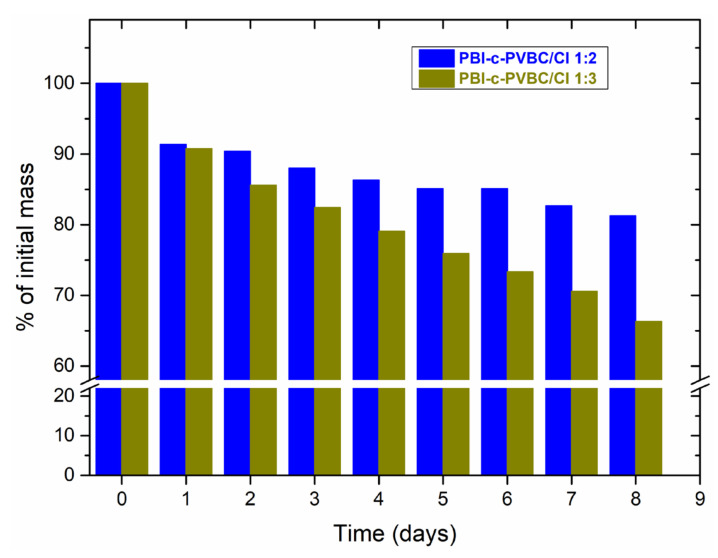
Loss of mass under accelerated durability test in oxidative media of PBI-c-PVBC/Cl membranes.

**Figure 9 membranes-10-00349-f009:**
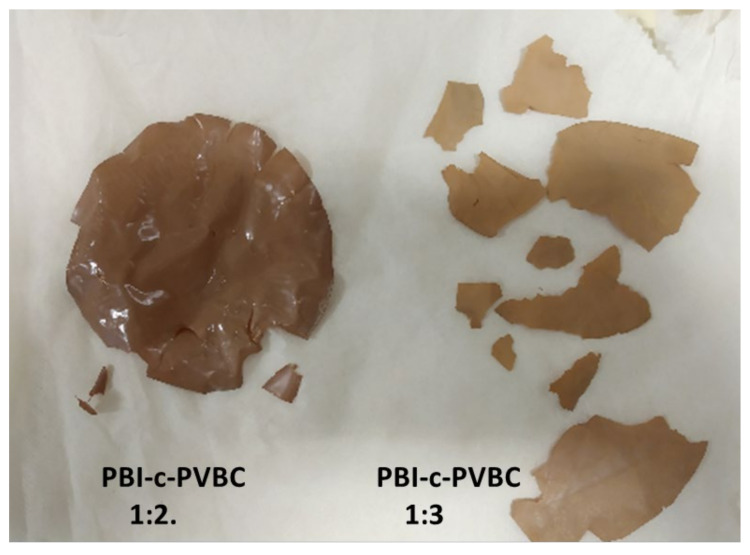
Visual appearance of the polybenzimidazole-c-PVBC membranes after the accelerated oxidative test.

**Figure 10 membranes-10-00349-f010:**
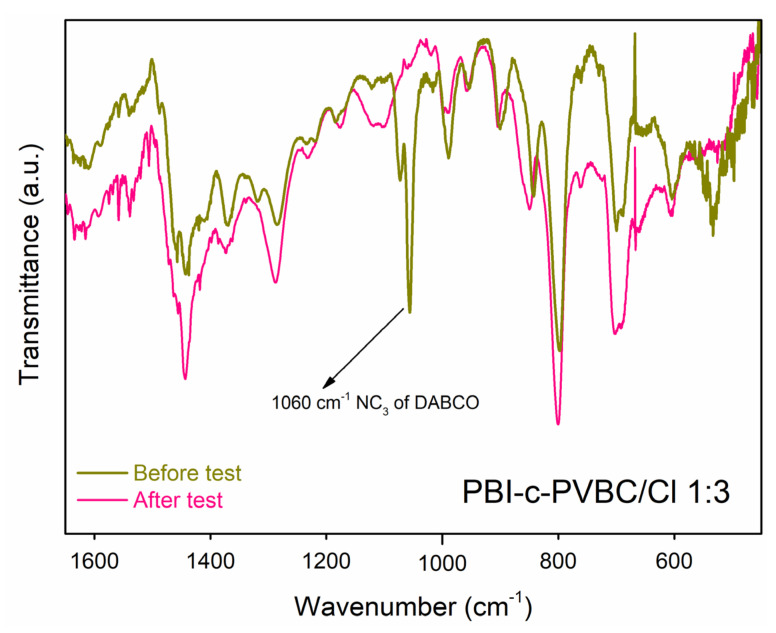
FT-IR spectra of the PBI-c-PVBC/Cl 1:3 membrane before and after the accelerated oxidative degradation durability test.

**Figure 11 membranes-10-00349-f011:**
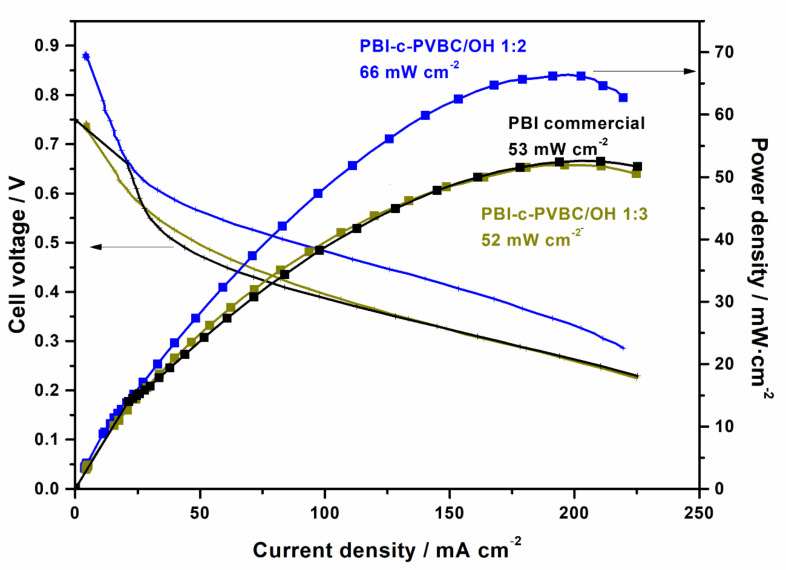
Direct ethanol fuel cells (DEFC) polarization (line) and power density (squares) curves using PBI-c-PVBC/OH membranes and compared to pristine PBI/KOH membrane. Commercial PBI result from [24].

**Figure 12 membranes-10-00349-f012:**
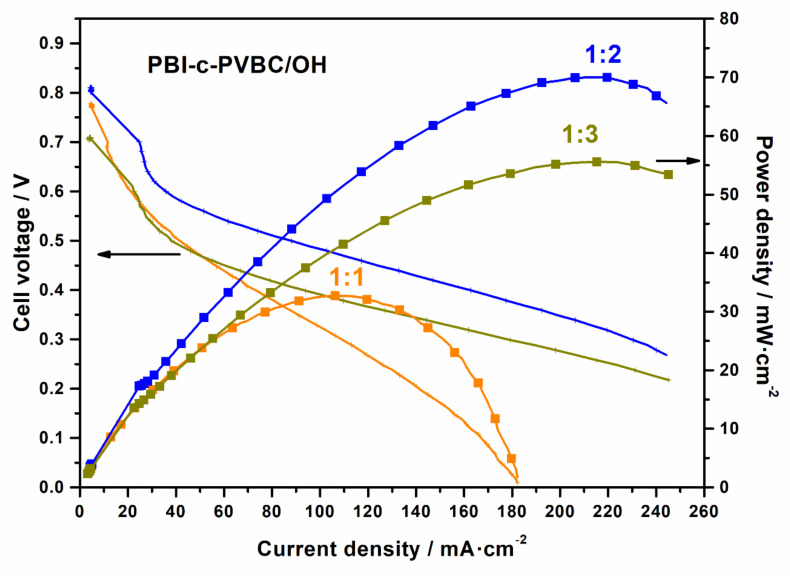
Fuel cell polarization (line) and power density (squares) curves for PBI-c-PVBC/OH 1:1 (orange), 1:2 (blue) and 1:3 (dark yellow) membranes. More details in Table 2.

**Table 1 membranes-10-00349-t001:** Conductivity and ion exchange capacity (IEC) values of the PBI-c-PVBC/OH membranes and variation of properties after the chemical stability test (immersion in 1M KOH solution at 60 °C for 20 days). (−) Reduction. (+) Increment. Data from [23].

	Conductivity 90 °C (mS·cm^−1^)	IEC (mmol·g^−1^)	IEC (%)	Water Uptake (%)	Vol. Swelling (%)
PBI-c-PVBC/OH 1:2	30	1.74	(−)10	(+)20	(+)18
PBI-c-PVBC/OH 1:3	45	1.97	(−)16	(+)11	(+)30

**Table 2 membranes-10-00349-t002:** PBI-c-PVBC/OH membranes of different molar ratios and the obtained DEFC average peak power density under various O_2_ backpressures.

Membrane	Thickness (µm)	Average Power Density (mW cm^−2^)	Temperature (°C)	O_2_ Backpressure (bar)
PBI-c-PVBC/OH 1:1	36	32	90	3
PBI-c-PVBC/OH 1:2	32	70	90	2
PBI-c-PVBC/OH 1:3	38	49	90	1

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
