# Peer review of "Application of Crosslinked Polybenzimidazole-Poly(Vinyl Benzyl Chloride) Anion Exchange Membranes in Direct Ethanol Fuel Cells"

_membranes, 2020, doi:10.3390/membranes10110349_

Round 1
Reviewer 1 Report
Summary:
This research entitled “Application of crosslinked polybenzimidazole-poly(vinyl benzyl chloride) anion exchange membranes in direct ethanol fuel cells” reports crosslinked membranes based on polybenzimidazole (PBI) and poly(vinyl benzyl chloride) (PVBC) by casting process for direct ethanol fuel cells. The performance of the resulting direct ethanol fuel cells with the PBI-c-PVBC/OH 1:2 membrane attains 66 mW cm-2 peak power density with the testing condition of 0.5 bar backpressure of pure O2 in the cathode and 1 mL min-1 KOH 2M EtOH 2 M aqueous solution in the anode. The value is about 25% higher than that of the cells with commercial PBI membranes
General comment:
In general, this submission gives a necessary improvement in electrochemical performance as compared to the reference. A few comments below are listed for the authors as a reference. Thank you.
Comments:
1. Experimental: The commercial carbon cloth seems to play an important role in the membrane-electrode assembly. Please provide the sources of the commercial carbon cloth. Please also give its basic physicochemical properties.
2. Results and Discussion: It is suggested to give some discussions about the porosity and permeability of the anion exchange membranes.
3. Results and Discussion: The mechanical strength of the PBI-c-PVBC/OH 1:2 membrane should be analyzed for a comparison to the commercial product.
4. Results and Discussion: Please clarify the meaning of “the after single cell tests” on line 302. For example, after how many single cell testing cycles or for how long in both cases of PBI-c-PVBC/OH 1:2 and 1:3 membranes.
Thank you for considering my reviewer comments. Hope this will help.
Author Response
Point 1: Experimental: The commercial carbon cloth seems to play an important role in the membrane-electrode assembly. Please provide the sources of the commercial carbon cloth. Please also give its basic physicochemical properties.

Response 1:
We thank Reviewer 1 for his/her contributions. We agree in the importance of giving the reference of the commercial carbon cloth employed in this work. Actually, the reference is already in the manuscript, but we understand the wording can be confusing. The GDL and MPL might be understood as two different components that we put together, when they are part of the same bought piece. Based on the reviewer suggestions, the sentence has been changed to:
“A typical membrane-electrode assembly (MEA) was prepared with commercial carbon cloth as gas-diffusion layer (GDL) with a carbon microporous layer on one side (MPL) (ELAT - LT1400, Fuel Cell Store, 454 µm and 63% porosity) over which the deposition of the anode and cathode catalyst layers were performed.”
Point 2: Results and Discussion: It is suggested to give some discussions about the porosity and permeability of the anion exchange membranes.
Response 2:
We agree on the importance of these parameters for the membrane's performance. In this work the porosity and permeability have not been directly measured, but they can be inferred from the shown fuel cell tests. For example, in the sentence “The higher OCP is related with a lower crossover of the reactants through the membrane from cathode to anode, exhibiting PBI-c-PVBC/OH 1:2 an improved barrier effect than PBI and PBI-c-PVBC/OH 1:3.” in line 438, in the discussion of the fuel cell results. If the porosity or permeability were too high, significant crossover of the reactants would occur and no high OCP values would be reached. To reflect it better, in the manuscript the previous sentence has been modified to:
“..exhibiting PBI-c-PVBC/OH 1:2 an improved barrier effect and therefore reduced porosity and permeability than PBI and PBI-c-PVBC/OH 1:3.”
Point 3: Results and Discussion: The mechanical strength of the PBI-c-PVBC/OH 1:2 membrane should be analyzed for a comparison to the commercial product.
Response 3:
We understand Reviewer 1 is referring to the commercial PBI membrane we are using as reference. The mechanical properties of commercial PBI membrane are described in detail in our previous article (Renew. Energy 127, (2018), 883-895, reference #24) and the maximum stress value of the doped membrane is included in the manuscript (Fig. 7), too. In the mechanical properties discussion (from line 350), the comparison with this commercial membrane is explained. The measurement is done on PBI-c-PVBC/Cl membranes and not directly for PBI-c-PVBC/OH ones, however, expected mechanical properties of the last are discussed in the end of the paragraph. We consider a better wording of the sentence should be given to highlight which membranes we are referring to, so the text in the manuscript has been modified accordingly:
“As these membranes exhibit high fuel cell performance, we consider that the obtained mechanical properties for PBI-c-PVBC/Cl membranes are adequate for final device applications. Membranes were immersed in KOH 1 M at room temperature before the fuel cell test to convert them to PBI-c-PVBC/OH. Although the humidification process on quaternized membranes is expected to alter their mechanical properties (decrease of maximum stress and increase of elongation) [30], it should not be an obstacle for their use in fuel cells according to our previous experience with PVA:polybenzimidazole membranes in DEFC tests.”
In the discussion of the fuel cell results, more details are given relating the mechanical properties of these membranes and their fuel cell performance, also comparing the commercial one. In conclusion, we consider the mechanical strength of the membranes compared with the commercial product is well discussed and the good mechanical properties of the PBI-c-PVBC/OH 1:2 membranes are sufficiently probed by the analysis of the mechanical properties of PBI-c-PVBC/Cl 1:2 and the performance in the fuel cell test.
In case Reviewer 1 is referring to other membranes like Tokuyama, we consider it is not relevant to compare with those ones since the chemical composition is too different. Being commercial PBI membranes available, they are a much better reference for the membranes under study in this work.
Point 4: Results and Discussion: Please clarify the meaning of “the after single cell tests” on line 302. For example, after how many single cell testing cycles or for how long in both cases of PBI-c-PVBC/OH 1:2 and 1:3 membranes.
Response 4:
We agree a clarification or better wording is needed in that sentence. The SEM/EDX study was performed in the membranes after the test was finished in the fuel cell, therefore, we have modified the text accordingly:
“The SEM microanalysis of the quaternized membranes used in MEA configuration after been tested in the fuel cell rendered highly relevant results.”
On each experiment different number of polarization curves and/or time was needed to obtain stable measurements so very specific details cannot be given.

Reviewer 2 Report
The present manuscript is to report on crosslinked membranes based on PBI and PVBC as anion exchange membrane, including the membrane properties and AEMFC performance. The experimental set strategy is reasonable, and the contents discussed are well described. Lacking is ion conductivity of the membrane and its stability study, the central interest of AEMFC research field. Fenton test is OK, but more appropriately the chemical stability under basic aqueous condition should be investigated.
In conclusion, I recommend to re-evaluate its value of publication, after additional experiment is completed.
Author Response
Response to Reviewer 2 Comments
Point 1: The present manuscript is to report on crosslinked membranes based on PBI and PVBC as anion exchange membrane, including the membrane properties and AEMFC performance. The experimental set strategy is reasonable, and the contents discussed are well described. Lacking is ion conductivity of the membrane and its stability study, the central interest of AEMFC research field. Fenton test is OK, but more appropriately the chemical stability under basic aqueous condition should be investigated.
Response 1:
We thank Reviewer 2 for his/her contribution. Although the objective of this research lies in the understanding of the membranes properties and quaternization process, as well as their application in direct ethanol fuel cells, we agree to highlight the conductivity properties of these membranes.
Therefore, beside the comments regarding the ionic conductivity of the membranes included in the discussion (paragraphs from 431- 441 and 456-461 lines), we include an additional table in the Supplementary Information, providing the necessary reference in the discussion (line 457). As this information was previously discussed in a previous article (Renew. Energy, 157, (2020), 71-82), we include this information in the SI section but not in the manuscript.
Table S1. Conductivity and IEC values of the PBI-c-PVBC/OH membranes and variation of properties after the chemical stability test (immersion in 1M KOH solution at 60 ºC during 20 days). (-) Reduction, (+) Increment. Data from [1].
|
Conductivity 90 oC (mS·cm-1) |
IEC (mmol·g-1) |
IEC (%) |
Water uptake (%) |
Volume swelling (%) |
PBI-c-PVBC/OH 1:2 |
30 |
1.74 |
(-) 10 |
(+) 20 |
(+) 18 |
PBI-c-PVBC/OH 1:3 |
45 |
1.97 |
(-) 16 |
(+) 11 |
(+) 30 |

Round 2
Reviewer 2 Report
The authors did put conductivities results in supporting information, but I recommend them to put them in the main manuscript. Plus, all experimental information for all results described in the supporting information should be given. After these revisions, the manuscript may deserve publication.
Author Response
Response to Reviewer 2 Comments
Point 1: The authors did put conductivities results in supporting information, but I recommend them to put them in the main manuscript. Plus, all experimental information for all results described in the supporting information should be given. After these revisions, the manuscript may deserve publication.

Response 1:
We thank Reviewer 2 for his/her contribution. Following your suggestion, the table has been included in the main manuscript and the experimental part has been completed, including the following description:
“The membrane swelling was calculated based on the thickness, length and width before and after immersion of the dry membrane in 1 M KOH aqueous solution, at room temperature for 48 h.
The IEC determination was performed by titration with aqueous 0.02M H2SO4 of 50 mL of 1M NaCl aqueous solution where the membrane sample was immersed during 24 h.
The through plane ionic conductivity measurements was determined by the impedance method (EIS), using an Autolab PGSTAT 30N coupled to a frequency response analyzer with a two-point technique.”
